# Linezolid Concentrations in Plasma and Subcutaneous Tissue are Reduced in Obese Patients, Resulting in a Higher Risk of Underdosing in Critically Ill Patients: A Controlled Clinical Pharmacokinetic Study

**DOI:** 10.3390/jcm9041067

**Published:** 2020-04-09

**Authors:** Philipp Simon, David Busse, David Petroff, Christoph Dorn, Lisa Ehmann, Sophie Hochstädt, Felix Girrbach, Arne Dietrich, Markus Zeitlinger, Frieder Kees, Charlotte Kloft, Hermann Wrigge

**Affiliations:** 1Department of Anaesthesiology and Intensive Care, University of Leipzig Medical Centre, 04103 Leipzig, Germany; sophie.hochstaedt@googlemail.com (S.H.); felixfrederic.girrbach@medizin.uni-leipzig.de (F.G.); hermann.wrigge@bergmannstrost.de (H.W.); 2Integrated Research and Treatment Centre (IFB) Adiposity Diseases, University of Leipzig, 04103 Leipzig, Germany; david.petroff@zks.uni-leipzig.de (D.P.); arne.dietrich@medizin.uni-leipzig.de (A.D.); 3Department of Clinical Pharmacy and Biochemistry, Institute of Pharmacy, Freie Universitaet Berlin, 12169 Berlin, Germany; david.busse@fu-berlin.de (D.B.); lisa.ehmann@fu-berlin.de (L.E.); charlotte.kloft@fu-berlin.de (C.K.); 4Graduate Research Training program PharMetrX, 12169 Berlin, Germany; 5Clinical Trial Centre Leipzig, University of Leipzig, 04107 Leipzig, Germany; 6Institute of Pharmacy, University of Regensburg, 93053 Regensburg, Germany; christoph.dorn@chemie.uni-regensburg.de; 7Department of Visceral, Transplant, Thoracic and Vascular Surgery, Division of Bariatric and Metabolic Surgery, University of Leipzig Medical Centre, 04103 Leipzig, Germany; 8Department of Clinical Pharmacology, Medical University of Vienna, 1090 Vienna, Austria; markus.zeitlinger@meduniwien.ac.at; 9Department of Pharmacology, University of Regensburg, 93053 Regensburg, Germany; frieder.kees@web.de; 10Department of Anaesthesiology, Intensive Care and Emergency Medicine, Pain Therapy, Bergmannstrost Hospital Halle, 06112 Halle, Germany

**Keywords:** antibiotic dosing, concentrations, linezolid, microdialysis, obesity, pharmacokinetics, soft tissue

## Abstract

**Background:** Linezolid is used for the treatment of soft tissue infections in critically ill patients. However, data for characterizing the pharmacokinetics (PK) and assessing whether effective concentrations are reached at the target site are lacking. We hypothesized that current dosing regimens do not lead to effective concentrations in the plasma and interstitial fluid (ISF) of subcutaneous tissue in obese patients. **Methods:** As a controlled clinical model, critically ill obese and non-obese patients undergoing intra-abdominal surgery received 600 mg linezolid as a single infusion. Concentrations in the plasma and microdialysate from the ISF of subcutaneous tissue were determined up to 8 h after dosing. Pharmacokinetic analysis was performed by non-compartmental methods. As a therapeutic target, we used *f*AUC/MIC > 80. **Results:** Fifteen obese (BMI: 48.7 ± 11.2 kg/m^2^) and 15 non-obese (23.9 ± 2.1 kg/m^2^) patients were analyzed. AUC_0–8_ in ISF decreased by −1.69 mg*h/L (95% CI: −2.59 to −0.79, *p* < 0.001) for every 10 kg increase in weight. PK in obese patients were characterized by lower maximal plasma concentrations (median 3.8 vs. 8.3 mg/L, *p* < 0.001) and a higher volume of distribution (41.0 vs. 30.8 L, *p* < 0.001), and the therapeutic target was not reached for MIC ≥ 1 mg/L in ISF and ≥ 2 mg/L in plasma. **Conclusions:** Increasing the weight led to a decrease of linezolid concentrations in the plasma and subcutaneous tissue. The current dosing regimen does not seem to produce sufficient concentrations to kill bacteria with MIC ≥ 2 mg/L, especially as empirical antimicrobial therapy in critically ill obese patients.

## 1. Introduction

Linezolid is used for the treatment of complicated skin and soft tissue infections with methicillin-resistant *Staphylococcus aureus* [1,2,3] and for toxic shock syndrome [4], and has become an important treatment option in empirical antimicrobial therapy in intensive care medicine, e.g., recommended in German guidelines for septic shock with a risk of methicillin-resistant *Staphylococcus aureus* (MRSA) infection [5]. In addition, linezolid is a frequently used antibiotic for the treatment of vancomycin-resistant *Enterococcus* (VRE), which is increasingly triggering serious infections in critically ill patients. For the success of an anti-infective therapy, as well as the prevention of the development of resistance, choices related to the right antibiotic, the duration of therapy, and the correct dosing regimen play a decisive role. Effective therapy requires sufficiently high antibiotic concentrations for a sufficient time period at the target site. Recent studies suggest that the current standard dosing regimen of 600 mg twice daily is insufficient for pathogens [6,7] such as MRSA [8]. Therefore, a discussion on the optimal dosing regimen for linezolid is currently underway [8,9,10]. Studies almost always use plasma concentrations as the basis for their analyses. However, target concentrations should not only include plasma, but above all, the site of bacterial infection, which is often the interstitial fluid (ISF) of the target tissue [11,12,13].

Obesity is widespread and is exhibiting an increasing incidence worldwide [14]. It is considered a risk factor for acquiring severe infections and the obese population has particularly severe problems once infections are acquired [15]. Therefore, adequate antibiotic therapy with sufficient effective concentrations is particularly important for obese patients. However, guideline recommendations for linezolid suggest weight-independent dosing [16]. The plasma volume does not differ between non-obese and obese patients nearly as much as the volume of soft tissue [17]. Given the similar plasma volumes, the dependence on the weight of plasma exposure cannot be expected to be as large as that for the ISF, i.e., for the target site. Only a few studies with a limited number of patients have focused on target site concentrations of linezolid, but without including obese patients [6,18,19,20,21] and without investigating the influence of obesity compared to a non-obese control group [22,23]. Consequently, there is a lack of data for characterizing the pharmacokinetics (PK) in this population and assessing whether effective linezolid concentrations are reached.

We hypothesized that the approved dosing regimen for linezolid does not lead to effective concentrations in the plasma and ISF of subcutaneous adipose tissue (one possible target site) in morbidly obese patients. To approach the problem with a clinically feasible model suitable for intensive care, but free from numerous influences, we accrued obese and non-obese patients undergoing elective abdominal surgery in a controlled setting and measured the free linezolid concentrations in soft tissue by microdialysis.

## 2. Materials and Methods

Patients were recruited for this prospective, parallel group, open-label, controlled single-center trial as part of a larger pharmacokinetic study at Leipzig University Hospital (German trial register number: DRKS00004776, www.who.int/ictrp/network/drks2/en/). Approval for the trial was granted by the Leipzig University ethics committee (121/13-ff) and the Federal Institute for Drugs and Medical Devices of Germany (BfArM), and was registered in the European Union clinical trials register (EudraCT number 2012-004383-22). All patients gave written informed consent prior to inclusion. After taking dropouts into account, a sample size of 15 per group was determined for the primary endpoint to find a correlation coefficient between the area under the concentration–time curve and weight of 0.52 with a power of 80%. The expectation for the correlation coefficient was taken from data presented in publications along with simulation studies [11,12]. Further details on the methods, design, and sample size calculations have previously been described in detail, including a CONSORT diagram for the whole study [24].

### 2.1. Study Population

Patients scheduled for elective abdominal surgery at Leipzig University Hospital were screened for eligibility. Inclusion criteria comprised age ≥ 18 years and body mass index (BMI) ≥ 35 kg/m^2^ for obese patients or between 18.5 and 30 kg/m² for non-obese patients. The non-obese patients were matched based on age (within five years) and sex, depending on the availability of sequentially encountered patients [24]. The main exclusion criteria were pregnancy or breastfeeding, a known allergic reaction to one of the investigated medications, and severe liver or severe kidney disease.

### 2.2. Intervention

Anesthesia was performed according to clinical standards and at the discretion of the anesthetist. The local clinical standard included either balanced anesthesia with propofol and sufentanil or remifentanil followed by desflurane/isoflurane, or total intravenous anesthesia. All patients received an infusion of crystalloid fluid, with or without additional vasopressors such as theodrenaline/cafedrine and/or norepinephrine, as required, to assure a mean arterial pressure > 60 mmHg. All patients received 1000 mg prophylaxis meropenem as a perioperative antibiotic and 600 mg linezolid as a clinical model, administered with a single 30 min infusion after the induction of anesthesia (60–30 min prior to incision) through an additional vein access procedure.

### 2.3. Blood and Microdialysate Samples

Two microdialysis catheters were placed in the interstitial fluid of subcutaneous adipose tissue of both upper arms (one in each arm) after skin disinfection and under sterile conditions, without using local anesthesia, ensuring that the complete length was subcutaneous, but without using ultrasound for the detection of anatomical landmarks. The catheters were placed 90 min before beginning anesthesia and the administration of the study drugs. The catheters were perfused with 0.9% saline at 2 µL/min. After equilibration (30 min), microdialysate was collected for 1 h and a baseline blood sample was taken. Additional blood samples were taken at 0.5 (end of infusion), 1, 2, 3, 4, 5, 6, and 8 h. Microdialysate samples were collected from 0–0.5, 0.5–1, 1–1.5, 1.5–2, 2–3, 3–4, 4–5, 5–6, 6–7, and 7–8 h [24]. The choice to sample for 8 h was based on a compromise between minimizing the burden to the patient, practicability, and ensuring rich sampling for various substances with different half-lives from a more extensive protocol (linezolid, meropenem, tigecycline, cefazolin, metronidazole, and piperacillin).

Blood samples were centrifuged within 30 min of sampling (3000 *g*, 4 °C, 10 min). Plasma and microdialysate samples were frozen immediately at −25 °C and then stored at −80 °C until analysis.

### 2.4. Microdialysis

Microdialysis was performed at a constant flow rate of 2 µL/min. To obtain the drug concentration in the ISF from microdialysate concentrations (C_µD_), the retrodialysis method was used [24,25]. Retroperfusate (C_RP_) and retrodialysate (C_RD_) drug concentrations were utilized to determine the in vivo relative recovery as follows: Relative recovery = (1 – C_RD_/C_RP_) × 100%. The concentration of the drug in the ISF was then given by C_ISF_ = C_µD_ × 100/relative recovery [24]. Due to the constant perfusate flow through the catheter and hence incomplete equilibrium at the membrane, only a fraction of the actual ISF drug concentration will be ‘recovered’ in the microdialysate, which is referred to as ‘relative recovery’ [26].

### 2.5. Study Visits and Data Collection

Patients were visited preoperatively, intraoperatively, and daily between postoperative day 1 and 7 or until discharge, as described in the visit schedule described elsewhere [24]. Study data were collected and managed using an Oracle-DBMS database with eResearch Network. Electronic data capture tools were employed at the Clinical Trials Centre of the University of Leipzig, Germany.

### 2.6. Drug Analysis

Linezolid was determined by a validated HPLC-UV method, as described previously [26,27]. Sample treatment for the determination of total linezolid included the deproteinization of plasma with perchloric acid, and microdialysate was directly injected into the column. The linearity of the method has been proven from 0.1 to 100 mg/L for plasma and from 0.1 to 300 mg/L for saline. Saline was used as a surrogate for microdialysate, as enough blank microdialysate was not available and a matrix effect is not observed when HPLC-UV is used. The coefficients of linear correlation (plasma/saline) were > 0.990/0.9991, the coefficients of variation for inter- and intra-assay precision (plasma/saline) at the lower limit of quantification (LLOQ) as determined by back-calculation were < 3%/< 1%, and the accuracy was 102.6%/100.7%, respectively. Based on in-process quality control (QC) samples, the intra- and inter-assay coefficient of variation and the in-accuracy (bias) was < 3%. Free linezolid in plasma was determined after ultrafiltration. The ultrafiltrate was injected directly [27]. The accuracy for free linezolid cannot be specified, as the protein binding in a particular plasma sample is not known. As the intra-assay variation was as low as 1% in preliminary experiments, QC samples (pooled plasma of healthy subjects spiked with 20, 4 or 0.8 mg/L linezolid) were analysed with each assay as single determinations. The unbound fraction in the QC samples (protein binding of linezolid is independent of the concentration) was 78.2 ± 2.0% (CV 2.6%).

### 2.7. Study Endpoints

The primary endpoint was the dependence of the area under the concentration–time curve from 0 to 8 h (AUC_0–8_) in the ISF on weight. Secondary outcomes were the weight dependence of the AUC_0–8_ in plasma (AUC_plasma_), and the weight dependence of the linezolid maximum concentration (C_max_), half-life (t_1/2_), clearance (CL), volume of distribution (V) at a steady state (V_ss_), and AUC_ISF_/AUC_plasma_.

We used the currently recommended target daily AUC of unbound concentrations divided by a minimum inhibitory concentration (*f*AUC_24_/MIC) > 80 [28,29] for an exploratory analysis and selected different relevant MIC values for linezolid (0.5–4 mg/L), with 4 mg/L defined as the linezolid-susceptible breakpoint for most Gram-positive isolates, including multiresistant pathogens [30]. As noted in [24], some endpoints foreseen in the protocol, such as wound infections, were not informative in our context and were thus only presented through summary statistics.

### 2.8. Pharmacokinetic and Statistical Analysis

ISF and plasma concentrations were analysed on a logarithmic scale and data from the two catheters were averaged. Non-compartmental pharmacokinetic analysis (NCA) was carried out using Phoenix WinNonlin 8.1 (Certara, Princeton, NJ, USA). The elimination rate constant λ_z_ was determined by log-linear regression in the elimination phase, which required at least three data points per patient and typically began by 4 h (plasma) or 4.5 h (ISF). The linear-up/log-down trapezoidal rule was used for the calculation of AUC. The extrapolation of AUC to infinity (AUC_∞_) was based on the last predicted concentration and the elimination constant (k_el_). The AUC for a daily dose of 1200 mg linezolid was calculated as AUC_24_ = 2 × AUC_∞_ after 600 mg linezolid, as determined in the present study. The estimation of AUC_24_ allowed us to relate our findings to the established target, *f*AUC_24_/MIC > 80, as mentioned above. The total body clearance (CL) and volume of distribution at a steady state (V_ss_) were calculated from plasma data. V_ss_ was found to be the product of the mean resistance time (MRT) with the CL. MRT is the ratio of the area under the first moment curve divided by AUC, both extrapolated to infinity, minus half the infusion time.

For evaluating the dependence of AUC on weight and other demographic data, linear models with AUC as the dependent variable and the covariates sex and weight or BMI were analysed using the software R (version 3.6.0, Vienna, Austria). A model containing both weight and BMI was not considered because of co-linearity. Pearson’s correlation was estimated, but robust methods were used in the presence of outliers [31,32], and Fisher’s Z transform was used for the confidence interval. Linear mixed-models were used for longitudinal data for each catheter separately with a random effects term allowing an intercept and slope for each patient. The obesity status, catheter, and time were taken as fixed effects, along with a term for the interaction between obesity status and time. Values are reported as the mean ± standard deviation or median (interquartile range). Tests were always two-sided and *p*-values < 0.05 were considered statistically significant.

## 3. Results

### 3.1. Participants

Patients were enrolled from December 2013 until March 2016. The patient flow (Figure 1—CONSORT flow diagram) demonstrates that all patients that underwent surgery were included in the analysis. Table 1 presents the patient characteristics. Non-obese patients were prospectively matched by sex and age (± five years). All patients in the obese group underwent bariatric surgery. In the non-obese group, 11 patients underwent gynecological operations, and the remaining four operations involved the stomach, liver, kidneys, and appendix.

### 3.2. Primary (ISF) and Secondary Study Endpoint (Plasma): Correlation of AUC_0–8_ with Weight

Plasma and interstitial fluid data were available from all patients, though in one patient, one of the two catheters slipped out and could not be analysed. The median (interquartile range (IQR)) value for relative recovery in ISF was 48% (34%, 58%). The AUC_0–8_ in the ISF changed by −1.69 mg*h/L (95% CI: −2.59 to −0.79, *p* < 0.001) for every 10 kg increase in weight and had a correlation coefficient of −0.61 (95% CI: −0.80 to −0.32, *p* < 0.001) (see Figure 2A). In the sensitivity analysis, the extreme value from a patient with a weight of 230 kg was removed and the correlation coefficient became −0.54, *p* = 0.002.

The free concentrations in plasma had a median (IQR) of 86% (84%, 88%) of the total concentrations. In plasma, *f*AUC_0-8_ changed by −2.2 mg*h/L (95% CI: −3.1 to −1.2, *p* < 0.001) for every 10 kg increase in weight and the correlation coefficient was −0.68 (95% CI: −0.84 to −0.43, *p* < 0.001) (Figure 2B). Correlations between AUC_0-8_ and BMI were very similar to those with weight and were −0.60 (95% CI: −0.79 to −0.31, *p* < 0.001) and −0.64 (95% CI: −0.81 to −0.36, *p* < 0.001) in ISF and plasma, respectively.

### 3.3. Analysis of ISF and Plasma Data as Functions of Time

Linear mixed-models showed that ISF concentrations in the obese group were 0.52 (95% CI: 0.39 to 0.70, *p* < 0.001) times those of the non-obese group, but declined at a rate that was 1.06 (95% 1.00 to 1.11, *p* = 0.037) times that of the non-obese group (see Figure 3A). The catheters on the right yielded concentrations 0.95 (95% CI: 0.91 to 0.99, *p* = 0.015) times those on the left.

Plasma concentrations in obese patients were 0.67 (95% CI: 0.58 to 0.78, *p* < 0.001) times those of the non-obese group and the rate of decline was similarly, but not significantly, higher, with a factor of 1.04 (95 % CI: 1.00 to 1.08, *p* = 0.066) compared to the non-obese group (see Figure 3B).

### 3.4. PK Parameters in ISF and Plasma and Statistical Analysis

Table 2 describes the relevant main PK parameters. The V_ss_ was significantly higher in the obese group (41.0 L) compared to the non-obese group (30.8 L, 95% CI for the difference is 6.9 to 21.3 L, *p* < 0.001). C_max_ and AUC_0–8_ were significantly lower in both plasma and interstitial fluid in the obese group (see Table 2 for estimates and *p*-values). The *f*AUC_0–8 ISF_/*f*AUC_0–8 plasma_ ratio was somewhat, though not statistically significantly, lower in the obese group (0.53 vs. 0.62, *p* = 0.080).

### 3.5. PK Relations to MIC Values for Susceptible Bacteria

Table 3 shows the *f*AUC_24_/MIC ratios for MICs of 0.5, 1, 2, and 4 mg/L. With an MIC ≥ 2 mg/L, *f*AUC_24_/MIC was lower than the target of 80 in both groups in plasma and ISF. In the obese group in ISF, this also occurred with an MIC of 1 mg/L. In contrast to unbound plasma, the ratio in ISF was significantly lower in the obese group (*p* = 0.033, see Table 3 for estimates and confidence intervals).

## 4. Discussion

The main findings of this study were that linezolid concentrations in soft tissue and plasma depended on the weight and therapeutic targets of *f*AUC_24_/MIC > 80 [28,29] were below accepted values in intensive care medicine, particularly for obese patients.

Our data expanded on other studies, which only performed measurements in plasma and/or lacked a non-obese control group [7,10,33]. Obesity leads to increased extracellular fluid and consequently, to an increased volume of distribution with lower plasma concentrations [33,34,35]. In our study, the increase was from 31 to 41 L. It should also be noted that other studies of non-obese patients showed comparable estimates of V (26.7–31.9 L) [36]. The slightly lower ratio of *f*AUC_0–8 ISF_/*f*AUC_0–8 plasma_ in obese patients suggests that target site exposure among the obese may not directly follow penetration in normal weight patients. This agrees with other studies showing differences in this penetration ratio between obese and non-obese groups [37]. Hence, our results corroborated that measurement at the target site is required to assess the adequacy of anti-infective exposure [13]. This finding may well be even more relevant in critically ill patients, where volumes of distribution are generally higher [38]. In critically ill patients, the phenomenon of augmented renal clearance is also described, which leads to an enhanced renal function and thus has an effect on the effective levels of many antibiotics in critically ill patients [39,40]. In contrast, altered renal clearance does not seem to have a relevant effect on the elimination of linezolid [41,42]. Renal function accounted for only 16% of the variance in linezolid clearance [10]. Therefore, the use of linezolid may be advisable, especially in critically ill patients with an altered renal function.

Recommendations of current guidelines [1,2,3,5] and the survival benefit of linezolid therapy compared to vancomycin in soft tissue infections [43] will lead to the more frequent use of linezolid, especially in intensive care medicine. We have confirmed the numerous plasma-based scientific results to date showing that the current dose of linezolid is sufficient in non-obese patients for a low MIC of 0.5 or 1 mg/L. This is also true in the ISF of the target tissue, although concentrations were lower. Moreover, our therapeutic target was not reached in plasma and ISF with an MIC above 2 mg/L for non-obese patients, as found in previous studies [6,7,10,44]. The differences between ISF and plasma concentrations were considerable. In the former, there was a clinically relevant difference between obese and non-obese groups, and in obese patients, the target was not even reached at the low MIC of 1 mg/L. Particularly in pathogens with higher MICs, such as multiresistant pathogens (2 or 4 mg/L) [30], our results indicated that the current therapy may be inadequate in all patients, especially in obese patients, and may lead to an increased development of resistance. This could be particularly relevant for empirical antimicrobial therapy with linezolid in severe skin and soft tissue infections, toxic shock syndrome, or septic shock in critically ill patients.

These findings imply the need for dosing intensification, as is currently being discussed [8,9,45], especially in obese patients. However, results of studies investigating dose adjustment in obese patients under clinical conditions do not yet exist. The fraction of enterococci resistant to linezolid is currently reported to be 1.6% worldwide [46] and the rate is increasing [47,48,49]. In this context, therapeutic drug monitoring is recommended [50,51].

The mean t_1/2_ of 3.4 h in plasma was somewhat shorter than what has been summarized by Stalker et al. [41], although it agrees well with other published values in the peri-operative setting, where increased diuresis after surgery is a factor that is likely to be relevant [20].

The total CL of linezolid did not depend on weight, which indicates that no relative adjustment of the maintenance dose solely due to the obesity status may be necessary. However, since this is a single-dose study, such extrapolations to a steady state after multiple dosing should be performed with caution. However, our data do permit statements regarding the loading dosage. Based on a population kinetic approach, a dosing schedule of 450 mg linezolid thrice daily has been suggested, in order to avoid high peak plasma concentrations during multiple dosing [10]. Based on the result of the present study, an alternative and practicable twice daily dosing schedule could include a loading dose of 600 mg or even 900 mg, accounting for the higher volume of distribution in obese patients and the slower diffusion into the tissue, followed by standard dosing of 600 mg linezolid twice daily, accounting for the similar clearance in both groups, and preferably administered as prolonged infusion, starting, e.g., after 8 h instead of 12 h. Regarding further dose adjustments during multiple dosing in critically ill patients, therapeutic drug monitoring is advised due to the high variability of linezolid plasma concentrations in these patients [27,50,51].

Post-hoc, exploratory nonlinear modeling of mixed effects with compartment methods has been performed to investigate additional hypothesis-generating questions, such as those relating to different dosing regimens. [52]. Moreover, different body size descriptors, such as the ideal body weight and lean body weight, could be useful for gaining an in-depth understanding of PK [53] and were considered in the exploratory paper mentioned above, whereas we complied with the foreseen protocol analysis based on weight. Using the methods employed in nonlinear modeling, Monte-Carlo simulations allowed us to explore the probability of target attainment in a variety of scenarios.

This study addresses the PK of linezolid in plasma, as well as at a subcutaneous target site, in morbidly obese patients and includes a control group of non-obese patients. In addition, it is the first controlled study with a larger sample size for target site measurements. However, this study had some limitations. The sampling schedule was limited to 8 h due to the clinical setting covering less than three half-lives. However, the extrapolation for AUC_24_ was 24% and thus only marginally in excess of the recommended 20% for bioequivalence studies. In ISF, the extrapolation was 33%. Estimates of clearance and the distribution volume also rely on extrapolated data. As both groups are likely to be affected approximately equally, the comparison of obese and non-obese patients should by quite robust. Some of the differences may also be attributable to the different type of surgery (e.g., laparoscopy).

Anesthesia may have an effect on the elimination of linezolid with expected reduced clearance due to a possible reduction of the cardiac output. This could only be investigated with a control group without anesthesia but was not the aim of the study. Infections can occur in different tissues and extrapolation of the results from the ISF of subcutaneous adipose tissue requires caution. Patients without infection were examined and measurements were taken in healthy tissue, but other authors have shown that in diabetic patients, infected tissue has the same linezolid concentrations as in healthy tissue [54,55]. However, extrapolation from this single-dose study in non-infected surgical patients to infected critically ill patients, where problems of underdosing are expected to be more pronounced, should be done with caution [6,23].

## 5. Conclusions

An increasing weight led to a clinically significant continuous decrease of linezolid concentrations in plasma, but also, more importantly, at the target site. Furthermore, the current dosing regimen of linezolid did not seem to lead to a sufficient concentration for empirical antibiotic therapy in intensive care medicine with an expected MIC of 2 or 4 mg/L in plasma or at the subcutaneous target site, particularly in obese critically ill patients. Our data suggest the need to measure linezolid concentrations in obese and above all, critically ill patients, especially at the target site, over a longer multiple-dosing treatment period.

## Figures and Tables

**Figure 1 jcm-09-01067-f001:**
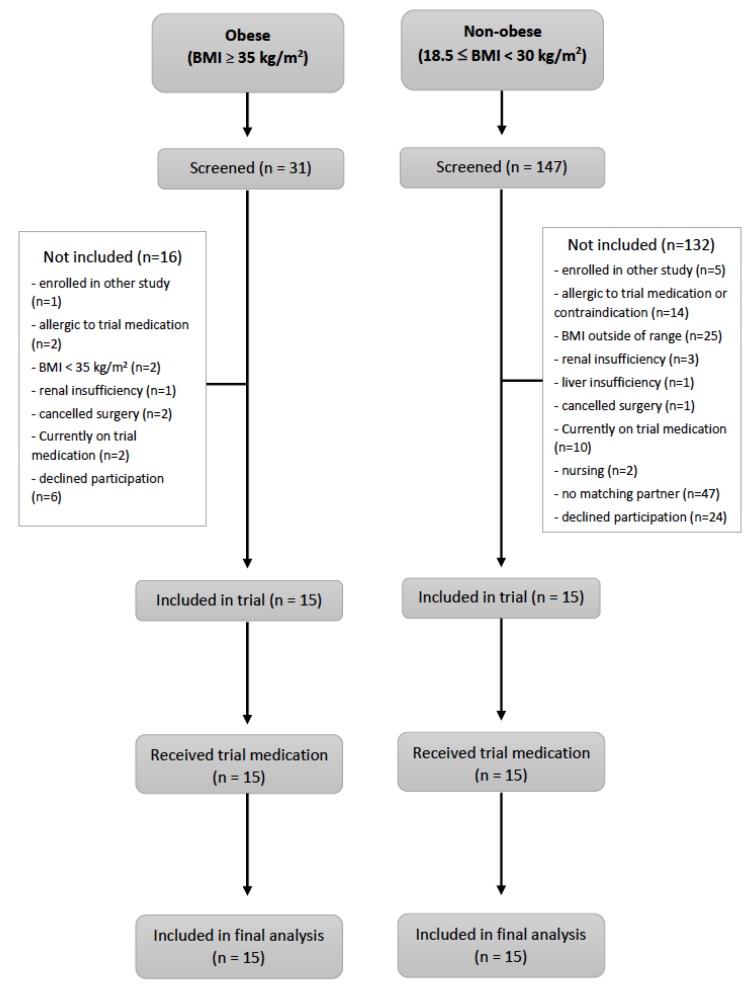
Flowchart of enrolment and outcomes. BMI = body mass index.

**Figure 2 jcm-09-01067-f002:**
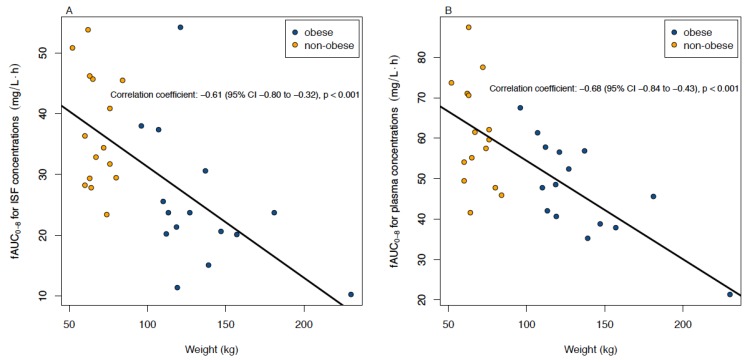
Correlation between the area under the concentration–time curve for unbound linezolid (*f*AUC_0–8_) (**A**) in the interstitial fluid of subcutaneous adipose tissue (ISF) and (**B**) in plasma and weight after single intravenous administration of 600 mg linezolid. *f*AUC = area under the concentration–time curve of unbound concentrations; CI = confidence interval.

**Figure 3 jcm-09-01067-f003:**
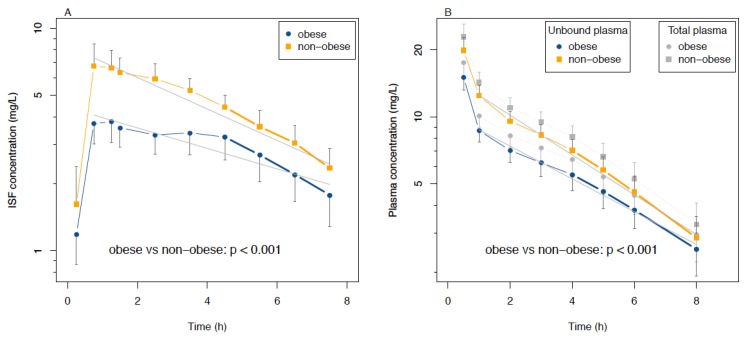
Concentration–time curves of linezolid (**A**) in interstitial fluid of subcutaneous adipose tissue (ISF) and (**B**) in plasma (total and unbound) after single administration of 600 mg linezolid intravenously. Whiskers indicate a 95% CI. The linear mixed-model was visualized by the solid (obese) and dashed lines (non-obese) of concentrations in ISF (Figure 3A) or unbound concentrations in plasma (Figure 3B) and formed the basis for the formal statistical comparison of the obese vs. non-obese groups. Note that half-lives were estimated by linear regression using the last 4–5 points in the curves and are not represented by the lines in the figure.

**Table 1 jcm-09-01067-t001:** Patient characteristics. Entries are the mean ± standard deviation or numbers (%). BMI = body mass index.

	Obese (*n* = 15)	Non-Obese (*n* = 15)
Number of females (*n* (%))	13 (87%)	13 (87%)
Age (years)	50.3 ± 9.5	49.5 ± 10.0
Weight (kg)	134.3 ± 34.3	67.9 ± 8.8
Height (m)	1.66 ± 0.07	1.68 ± 0.07
BMI (kg/m^2^)	48.7 ± 11.2	23.9 ± 2.1
Creatinine (µmol/L)	86.4 ± 27.2	75.3 ± 18.9
Type of surgery (*n* (%))		
laparoscopic	15 (100%)	3 (20%)
open	0 (0%)	12 (80%)
ASA classification (*n* (%))		
I	1	2
II	4	11
III	10	2

**Table 2 jcm-09-01067-t002:** Pharmacokinetic parameters for linezolid. Entries are the median [interquartile range], estimate (95% confidence interval), or count (%). The *p*-values refer to comparisons of the obese and non-obese groups. AUC = area under the concentration–time curve; CL = total body clearance; ISF = interstitial fluid of the subcutaneous adipose tissue; MRT = mean resistance time; V_ss_ = volume of distribution at a steady state; C_max_ = maximum concentration; t_1/2_ = half-life; T_max_ = time point of C_max_. Bold type indicates significant values.

	All Patients (*n* = 30)	Pearson’s Correlation Coefficient with Weight	Obese (*n* = 15)	Non-Obese (*n* = 15)	*p*-Value
C_max_ (mg/L)					
ISF	5.5 [3.8, 8.5]	**−0.70 (−0.85 to −0.45)**	3.8 [3.2, 5.3]	8.3 [5.7, 10.2]	**<0.001**
Plasma (total)	21.9 [17.2, 24.4]	**−0.63 (−0.81 to −0.36)**	19.2 [16.1, 22.1]	24.4 [18.9, 26.9]	**0.013**
Plasma (unbound)	18.5 [14.4, 21.1]	**−0.66 (−0.82 to −0.39)**	15.9 [13.3, 18.7]	21.1 [16.6, 23.3]	**0.007**
t_1/2_ (hours)					
ISF	3.9 [2.8, 4.8]	0.11 (−0.26 to 0.45)	3.9 [2.7, 5.1]	3.8 [3.0, 4.2]	0.664
Plasma **(total and unbound)**	3.4 [2.6, 4.4]	**0.52 (0.20 to 0.74)**	3.8 [3.0, 4.8]	3.0 [2.5, 3.6]	0.088
V_ss_ (L), plasma	37.7 [31.3, 44.6]	**0.87 (0.75 to 0.94)**	41.0 [38.4, 57.9]	30.8 [27.9, 35.5]	**<0.001**
CL (L/h), plasma	7.2 [6.3, 9.3]	0.21 (−0.16 to 0.53)	8.2 [6.2, 10.8]	7.0 [6.4, 8.2]	0.170
MRT (h), plasma	5.0 [3.8, 6.3]	**0.47 (0.13 to 0. 71)**	5.7 [4.5, 6.9]	4.4 [3.8, 5.4]	0.079
T_max_, ISF 30–60/60–90/>90 min (number of patients)	17/3/10	—	7/3/5	10/0/5	0.252
AUC_0-8_, (mg*h/L)					
ISF	29.4 [23.5, 37.8]	**−0.61 (−0.80 to −0.32)**	23.7 [20.2, 28.1]	34.4 [29.4, 45.5]	**0.004**
Plasma **(total)**	63.4 [52.8, 71.1]	**−0.66 (−0.82 to −0.39)**	57.9 [45.4, 65.6]	67.8 [61.5, 80.5]	**0.011**
Plasma **(unbound)**	54.5 [45.6, 61.5]	**−0.68 (−0.84 to −0.43)**	47.7 [39.7, 56.7]	59.6 [51.7, 70.8]	**0.005**
AUC_0–8 ISF_/*f*AUC_0–8 plasma_	0.53 [0.46, 0.65]	−0.28 (−0.58 to 0.09)	0.53 [0.45, 0.55]	0.62 [0.53, 0.68]	0.080

**Table 3 jcm-09-01067-t003:** Linezolid exposure in the interstitial fluid of subcutaneous adipose tissue and plasma relative to clinically relevant MIC values. Entries are the median [interquartile range] or estimate (95% confidence interval). The *p*-values refer to comparisons of the obese and non-obese groups. *f*AUC = area under the concentration–time curve of unbound concentrations; ISF = interstitial space fluid; MIC = minimal inhibitory concentration.

		All Patients (*n* = 30)	Pearson’s Correlation Coefficient with Weight	Obese (*n* = 15)	Non-Obese (*n* = 15)	*p*-Value
	MIC (mg/L)					
*f*AUC_24_/MIC						
ISF	0.5	176.8 [125.4, 218.2]	−0.47 (−0.71 to −0.13)	140.0 [105.6, 207.0]	182.4 [165.4, 239.6]	0.033
	1	88.4 [62.7, 109.1]		70.0 [52.8, 103.5]	91.2 [82.7, 119.8]	
	2	44.2 [31.3, 54.6]		35.0 [26.4, 51.7]	45.6 [41.3, 59.9]	
	4	22.1 [15.7, 27.3]		17.5 [13.2, 25.9]	22.8 [20.7, 30.0]	
plasma	0.5	286.6 [220.8, 324.0]	−0.42 (−0.68 to −0.08)	237.6 [199.8, 321.8]	289.4 [250.0, 324.0]	0.175
	1	143.3 [110.4, 162.0]		118.8 [99.9, 160.9]	144.7 [125.0, 162.0]	
	2	71.7 [55.2, 81.0]		59.4 [49.9, 80.5]	72.4 [62.5, 81.0]	
	4	35.8 [27.6, 40.4]		29.7 [25.0, 40.2]	36.2 [31.3, 40.5]	

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
