# Peer review of "Linezolid Concentrations in Plasma and Subcutaneous Tissue are Reduced in Obese Patients, Resulting in a Higher Risk of Underdosing in Critically Ill Patients: A Controlled Clinical Pharmacokinetic Study"

_jcm, 2020, doi:10.3390/jcm9041067_

Round 1

Reviewer 1 Report

This manuscript describes a novel study of linezolid pharmacokinetics in obese and non-obese patients, using plasma and interstitial fluid concentrations, the latter ascertained from microdialysate sampling. The work is well described and general well presented. Queries and comments that could improve the manuscript are provided below.

  1. The abstract may require revision to reflect changes in the manuscript.
  2. The authors’ message in the statement on lines 62-64 is unclear: “Plasma volume does not differ between non-obese and obese patients nearly as much as the volume of soft tissue [16]. Hence, plasma concentrations cannot be expected to depend on weight in the same way as those at the target site.” Could the authors re-phrase to clarify their message?
  3. The authors cite only one reference in the statement on line 65: “However, there are no recommendations for weight-dependent doses of linezolid [17]”. Other references in the manuscript (e.g., #7, #10, #31) have investigated the pharmacokinetics of linezolid in obesity and Cojutti et al. (#10) have suggested 450 mg every 8 h in obese patients. Furthermore, Hamilton et al. (J Antimicrob Chemother 2013; 68: 666-673) have suggested up to 600 mg every 8 h in obese patients. The authors should consider further elaboration on this point.
  4. The sampling schedule (lines 112-114) extends to only 8 hours, even though the established half-life of linezolid is in the order of 5-6 hours and sampling should normally extend to at least three half-lives in pharmacokinetic studies. Based on the authors reported half-life of 3-4 hours and LOQ of 0.1 mg/L, this reviewer estimates that sampling to 24 hours should have been plausible. The authors should address this matter.
  5. The LOQ should be confirmed with evidence of the assay coefficient of variation for both plasma and ISF.
  6. Further clarity on the AUC(0-24) is required in the manuscript. Although stated in line 164 that AUC(0-24) is 2 x AUC(0-∞), the rationale should be clarified and reiterated elsewhere (e.g., Table 2).
  7. Body size descriptors have not been reported but would appear to be an important element in this study (as considered by Bhalodi et al. #31). Whilst the mean IBW is likely to be similar in the two groups (referring to Table 1), adjusted BW and lean BW (see Han et al. Clin Pharmacol Ther 2007; 82: 505-508) may be in the order of 50% higher in the obese patients.
  8. Figure 2 is a very nice indicator of the rationale for higher doses of linezolid in obesity and could be used to justify specific recommendations.
  9. The data in Figure 3 suggest that the authors could have modelled distribution into the ISF, since the concentrations appear to be sustained for several hours after the dose. These data also show the limitation of sampling for only 8 h after the dose. This reviewer notes the discussion lines 277-279 refer to a manuscript under review – perhaps the authors could clarify why this work has been separated for publication in a different forum.
  10. It is unclear why data in table 2 are reported as median?
  11. The layout of table 2 is confusing; for example, it is not initially clear that the second and third lines are Cmax data.
  12. Please outline how Vss has been determined and include MRT in the table.
  13. In addition to reporting AUC(0-8), the AUC(0-∞) and the extrapolated AUC (as a proportion of total) should be provided. Comment on the extrapolated AUC would be required.
  14. It is not surprising that the AUC is lower in obesity and therefore leads to higher Vss and CL. However, whilst total V may be considered as the basis for linezolid loading doses, it is very unlikely that total CL would be used for linezolid maintenance dose calculations. Indeed, based on reports such as Green & Duffull (Br J Clin Pharmacol 2004; 58: 119-133), total BW or adjusted BW could be the best descriptor for V and lean BW may be preferable for CL. This reviewer estimates from the data in table 2 that Vss (L/kg) and CL (L/h/kg) would not be substantially different if normalised according to ABW or LBW. The calculated doses would be some 50% higher in the obese patients and should provide similar ISF AUC(0-8). The authors should further consider this point.
  15. Further to the previous point, the results and discussion would benefit from a dosing recommendation in obesity, not least to address the authors’ comment on line 65.
  16. In line 284 the proportion of extrapolated AUC should be specified.

Author Response

Point-by-point response to the reviewers’ comments to JCM- 747318
“Linezolid concentrations in plasma and subcutaneous tissue are reduced in obese patients resulting in a higher risk of underdosing in critically ill patients – a controlled clinical pharmacokinetic study”
by Simon et al.

Thank you very much for giving us again the opportunity to revise our manuscript entitled „Linezolid concentrations in plasma and subcutaneous tissue are reduced in obese patients resulting in a higher risk of underdosing in critically ill patients – a controlled clinical pharmacokinetic study”.

We thank the editors and reviewers for their criticism. We have discussed the points of criticism in detail and hope very much that after further revision you will come to the conclusion that the manuscript is suitable for publication Journal of Clinical Medicine.

Our responses to the reviewers’ and editors’ comments below are highlighted in italic, our changes to the manuscript (insertions and deletions) are indicated in red (deletions are crossed through). The line numbers refer to those in the revised manuscript with highlighted changes.

REVIEWER 1:

This manuscript describes a novel study of linezolid pharmacokinetics in obese and non-obese patients, using plasma and interstitial fluid concentrations, the latter ascertained from microdialysate sampling. The work is well described and general well presented. Queries and comments that could improve the manuscript are provided below.

We thank the reviewer for the overall positive feedback and the good suggestions.

  1. The abstract may require revision to reflect changes in the manuscript.

The abstract has been adapted accordingly and as per request by the Editor.

  1. The authors’ message in the statement on lines 62-64 is unclear: “Plasma volume does not differ between non-obese and obese patients nearly as much as the volume of soft tissue [16]. Hence, plasma concentrations cannot be expected to depend on weight in the same way as those at the target site.” Could the authors re-phrase to clarify their message?

We have rewritten this sentence to make it easier to understand (line 75-76).

  1. The authors cite only one reference in the statement on line 65: “However, there are no recommendations for weight-dependent doses of linezolid [17]”. Other references in the manuscript (e.g., #7, #10, #31) have investigated the pharmacokinetics of linezolid in obesity and Cojutti et al. (#10) have suggested 450 mg every 8 h in obese patients. Furthermore, Hamilton et al. (J Antimicrob Chemother 2013; 68: 666-673) have suggested up to 600 mg every 8 h in obese patients. The authors should consider further elaboration on this point.

This was a misunderstanding. We did not mean to imply that research has not broached the topic. We have changed the sentence to: „However, guideline recommendations for linezolid do not account for weight.“ (line 73-74)

  1. The sampling schedule (lines 112-114) extends to only 8 hours, even though the established half-life of linezolid is in the order of 5-6 hours and sampling should normally extend to at least three half-lives in pharmacokinetic studies. Based on the authors reported half-life of 3-4 hours and LOQ of 0.1 mg/L, this reviewer estimates that sampling to 24 hours should have been plausible. The authors should address this matter.

We agree with you, the other reviewer and the Editor that this point requires more attention and have added additional passages to the manuscript. We present arguments below suggesting that extrapolation may be quite good however.

Patient burden is a limiting factor for collecting ISF samples. However, the blood sampling could have covered 11-12 hours and this will be considered in future studies. A 12 hour blood sampling schedule in the clinical setting would be acceptable considering a terminal half-life of <5 hours established in a study with 24h sampling schedule (Stalker DJ, Jungbluth GL, Hopkins NK, Batts DH. Pharmacokinetics and tolerance of single- and multiple-dose oral or intravenous linezolid, an oxazolidinone antibiotic, in healthy volunteers. J Antimicrob Chemother 2003;51(5):1239–46.) Moreover, due to the strictly followed rich sampling schedule and the precise HPLC method, extrapolation over the last measured concentration was reliable - in plasma. The concentration time course in ISF was more variable. Also because of that, a non-compartmental analysis was preferred. 

We can certainly understand this concern. Note however that for plasma, the extrapolation of 24% for AUC is only marginally larger than the “allowed” 20%. Throughout the manuscript we have reduced emphasis on results relying on extrapolation and are more explicit in the methods about the sampling schedule (lines 133-136) and have added a passage to the discussion to help the reader assess this limitation (lines 365-368).

The guidance documents in question are intended to be applied in drug development, i.e. if the knowledge about a new drug is rather scarce. Our data analysis involves the long approved antibiotic linezolid with well characterised pharamcokinetics in plasma and non-obese. Our confidence that our sampling schedule (of 8 h) captures the 'true' terminal phase is based on the comparison to previous analysis: As many others, we see a biphasic PK profile and a start of the terminal phase around ~1-2 h. Hence, the assumption that we sampled in the terminal elimination phase and thus determined the terminal elimination rate constant is highly plausible especially considering the large number of samples in this phase for our log-linear regression.

It may also be of interest to note that similar sampling schedules can be found in the literature, e.g.  Thallinger C, Buerger C, Plock N, et al. Effect of severity of sepsis on tissue concentrations of linezolid. J Antimicrob Chemother, 2008 ;61(1):173–6. 

A further assumption is that the terminal phase extends beyond the last observed concentration, which might not always be a good assumption, especially when PK of a new drug is not well characterised. However, the PK of plasma/serum linezolid is very well-characterised in various patient populations and exclusively up to two exponential phases have been shown (compare e.g. Figure 1 in Dehghanyar P et al. Penetration of Linezolid into Soft Tissues of Healthy Volunteers after Single and Multiple Doses. Antimicrob Agents Chemother 2005). Thus it can be assumed that the terminal phase (as the second exponential phase) has been adequately described and AUCo-∞ is warranted.

Still, throughout the manuscript we have reduced emphasis on results relying on extrapolation and are more explicit in the methods about the sampling schedule (lines 133-136) and have added a passage to the discussion to help the reader assess this limitation (lines 365-368).

  1. The LOQ should be confirmed with evidence of the assay coefficient of variation for both plasma and ISF.

We have modified the section “Drug analysis” to address this (lines 163-169).

  1. Further clarity on the AUC(0-24) is required in the manuscript. Although stated in line 164 that AUC(0-24) is 2 x AUC(0-∞), the rationale should be clarified and reiterated elsewhere (e.g., Table 2).

We have explained AUC(0-24) more clearly, and removed it from Table 2, where it was superfluous, and have expanded on the rationale for using it in Table 3 (lines 201-202).

  1. Body size descriptors have not been reported but would appear to be an important element in this study (as considered by Bhalodi et al. #31). Whilst the mean IBW is likely to be similar in the two groups (referring to Table 1), adjusted BW and lean BW (see Han et al. Clin Pharmacol Ther 2007; 82: 505-508) may be in the order of 50% higher in the obese patients.

As to be expected, IBW is indeed similar in both groups (58 vs 60 kg) whereas ABW (88 vs 68 kg) and LBW (61 vs 44 kg) are about 30-40% higher in the obese group. In our Discussion, we have added a passage on body size descriptors, noting that they also play a role in our paper using nonlinear mixed modelling. (lines 357-360). We also discuss this point below regarding clearance.

  1. Figure 2 is a very nice indicator of the rationale for higher doses of linezolid in obesity and could be used to justify specific recommendations.

Thank you. Implicitly the results of the correlation analysis are used in the discussion, but we have not specifically emphasized Figure 2 again.

  1. The data in Figure 3 suggest that the authors could have modelled distribution into the ISF, since the concentrations appear to be sustained for several hours after the dose. These data also show the limitation of sampling for only 8 h after the dose. This reviewer notes the discussion lines 277-279 refer to a manuscript under review – perhaps the authors could clarify why this work has been separated for publication in a different forum.

To avoid misunderstandings, we have added a sentence to the caption in Fig. 3 explaining that half-lives were estimated with the terminal points in the curves. The choice of presenting the current analysis separately from the nonlinear mixed effects model with compartmental methods and Monte-Carlo (MC) simulations was intended to distinguish clearly between the foreseen protocol analysis and a post-hoc exploratory one. Such an analysis provides important insights, but is based on extrapolation from the data and does not specifically address the difference between obese and normal weight patients.

Moreover, the detailed description of the latter would have been beyond the scope of this manuscript. Especially given the substantial lack of data in the understudied (morbidly) obese population, we intend to cover both aspects of the study in depth (standardised characterisation of PK and explorative MC simulations of alternative dosing regimens).

  1. It is unclear why data in table 2 are reported as median?

Many of the variables presented in Table 2 have a skewed distribution (e.g. AUC0-24 for non-obese with a median of 91.2 and an IQR of 82.6 – 199.8). For such variables, the presentation was deemed more appropriate and informative than mean with SD. For those with a roughly normal distribution, the information from median and IQR contains almost the same information as mean and 1.3*SD.

  1. The layout of table 2 is confusing; for example, it is not initially clear that the second and third lines are Cmax data.

We have modified the layout as suggested.

  1. Please outline how Vss has been determined and include MRT in the table.

Vss is now defined in the Methods section (lines 202-204) and we added MRT to Table 2.

  1. In addition to reporting AUC(0-8), the AUC(0-∞) and the extrapolated AUC (as a proportion of total) should be provided. Comment on the extrapolated AUC would be required.

We have modified Table 2 and added a comment about extrapolated AUC in the discussion (lines 365-368).

  1. It is not surprising that the AUC is lower in obesity and therefore leads to higher Vss and CL. However, whilst total V may be considered as the basis for linezolid loading doses, it is very unlikely that total CL would be used for linezolid maintenance dose calculations. Indeed, based on reports such as Green & Duffull (Br J Clin Pharmacol 2004; 58: 119-133), total BW or adjusted BW could be the best descriptor for V and lean BW may be preferable for CL. This reviewer estimates from the data in table 2 that Vss (L/kg) and CL (L/h/kg) would not be substantially different if normalised according to ABW or LBW. The calculated doses would be some 50% higher in the obese patients and should provide similar ISF AUC(0-8). The authors should further consider this point.

We thank the reviewer for this very important hint, which helped us to identify a weak point in our analysis. We have now incorporated so-called robust analysis methods (see e.g. R. A. Maronna and V. J. Yohai (1995) The Behavior of the Stahel-Donoho Robust MultivariateEstimator.Journal of the American Statistical Association90(429), 330–341) in the presence of outliers. As a result, we see no correlation between total CL and weight. One single exceptionally heavy patient (BW 230 kg, BMI 81.5 kg/m²) affected the analysis disporportionately. The lack of correlation suggests that no adjustment of the maintenance dose in obese patients (in relation to non-obese patients) may be necessary. However, since this is a single dose study, such extrapolations to steady state after multiple dosing should be approached with caution.

On the contrary, statements deriving from our data regarding the initial dose are appropriate. The slighty higher calculated V in obese vs. non-obese patients (41 L vs 31 L) and the corresponding lower peak concentrations in plasma (19 mg/L vs 24 mg/L) and in ISF (3.8 mg/L vs. 8.3 mg/L) indicate that a loading dose of 600 mg linezolid as short infusion followed by a prolonged infusion of 600 mg linezolid after 8 hours may be appropriate, practical and still safe regarding toxicity (REF 10: Cojutti 2018). Regarding further dose adjustments during multiple dosing/long term therapy in ICU patients, a therapeutic drug monitoring is advised due to the high variability of linezolid plasma concentrations in critically ill patients (REF 27: Töpper 2016).

  1. Further to the previous point, the results and discussion would benefit from a dosing recommendation in obesity, not least to address the authors’ comment on line 65.

As a final goal, we agree that a dosing recommendation is necessary, but feel we cannot yet provide one on the basis of our data. We intend to provide the basis for future clinical studies in critically ill obese patients, hence the relevance in this special issue.

  1. In line 284 the proportion of extrapolated AUC should be specified.

We have now provided this proportion (line 366-367).

Reviewer 2 Report

  • Sampling schedule should be reconsidered. Although the dosage interval is every 12h samples only were taken until 8h. You must justify the chosen times. The last time should have been later because their extrapolated AUC is greater than 20% of total AUC.  This indicates that more sampling is needed for an accurate estimate of the elimination constant and the AUC. Therefore, these results are unreliable.
  • If the AUC and the elimination constant are not well estimated, CL and V calculation will not be well estimated either.
  • The differences in V and half-life observed versus published data reinforce the idea that sampling time may not be adequate.
  • The groups are not comparable in some aspects: 100% laparoscopies in the obese group versus 20% in the non-obese group. This aspect should be discussed.
  • This study was carried out after single dose without being in the steady state. We can see a higher V in obese patients versus non-obese patients. This data could provide information related to the loading dose required in this group of patients but not so much with the maintenance dose. In this sense I have to highlight:
  1. Although the V is higher in obese group, it is similar to published data in other groups of patients while the V in non-obese ones is lower that was has been published.
  2. The maintenance dose is more related to the clearance of linezolid.
  3. There is no information in the text about patients’ renal function. An adequate estimation of renal function by calculating creatinine clearance measured in urine is critical to interpreting linezolid clearance.
  4. In this study linezolid is used as a single dose in prophylaxis. I understand that's why the maintenance dose hasn't been studied. But the usual use of linezolid is not in this context. It would be interesting to analyze the factors that may condition the clearance of linezolid in order to estimate the necessary maintenance dose in this group.
  • Why has a non-compartmental model been selected instead of a compartmental one? Justify.
  • Instead of estimating the AUC/MIC value for each patient, a better approach would be to calculate the probability of target  attainment  using Monte Carlo simulation. This issue should be considered.

Author Response

Point-by-point response to the reviewers’ comments to JCM- 747318
“Linezolid concentrations in plasma and subcutaneous tissue are reduced in obese patients resulting in a higher risk of underdosing in critically ill patients – a controlled clinical pharmacokinetic study”
by Simon et al.

Thank you very much for giving us again the opportunity to revise our manuscript entitled „Linezolid concentrations in plasma and subcutaneous tissue are reduced in obese patients resulting in a higher risk of underdosing in critically ill patients – a controlled clinical pharmacokinetic study”.We thank the editors and reviewers for their criticism. We have discussed the points of criticism in detail and hope very much that after further revision you will come to the conclusion that the manuscript is suitable for publication Journal of Clinical Medicine.

Our responses to the reviewers’ and editors’ comments below are highlighted in italic, our changes to the manuscript (insertions and deletions) are indicated in red (deletions are crossed through). The line numbers refer to those in the revised manuscript with highlighted changes.

REVIEWER 2:

  • Sampling schedule should be reconsidered. Although the dosage interval is every 12h samples only were taken until 8h. You must justify the chosen times. The last time should have been later because their extrapolated AUC is greater than 20% of total AUC.  This indicates that more sampling is needed for an accurate estimate of the elimination constant and the AUC. Therefore, these results are unreliable.

This has been pointed out by both reviewers and the Editor and this concern is understandable considering the 20% extrapolation limit, though we point out that in plasma, our extrapolation is 24% (lines 366-367).

We now outline in the methods section lines 133-136 how we arrived at these sampling times. Still, we believe that a reliable extrapolation of AUC is achieved: AUClast-∞ is calculated based on the last predicted concentration and the elimination rate constant λz.

λz is calculated via log-linear regression from the terminal phase, which is well characterised in this study, owing to the dense sampling (see also our answer 6).

A further assumption is that the terminal phase extends beyond the last observed concentration, which might not always be a good assumption, especially when PK of a new drug is not well characterised. However, the PK of plasma/serum linezolid is very well-characterised in various patient populations and exclusively up to two exponential phases have been shown (compare e.g. Figure 1 in Dehghanyar P et al. Penetration of Linezolid into Soft Tissues of Healthy Volunteers after Single and Multiple Doses. Antimicrob Agents Chemother 2005). Thus it can be assumed that the terminal phase (as the second exponential phase) has been adequately described and AUCo-∞ is warranted.

 Still, throughout the manuscript we have reduced emphasis on results relying on extrapolation and are more explicit in the methods about the sampling schedule (lines 133-136) and have added a passage to the discussion to help the reader assess this limitation (lines 365-368).

  • If the AUC and the elimination constant are not well estimated, CL and V calculation will not be well estimated either.

Your point is well taken and we have included this in the limitations section (line 368).

  • The differences in V and half-life observed versus published data reinforce the idea that sampling time may not be adequate.

We note in the paper that the agreement with some published material is quite good, but with others less so. Of course, extrapolation could indeed explain some of the differences. We further elaborate on this point in our answer 5.

  • The groups are not comparable in some aspects: 100% laparoscopies in the obese group versus 20% in the non-obese group. This aspect should be discussed.

We have included this secondary aspect in the limitations so that the reader can take it into account in the interpretation. This is the only relevant difference between the groups and we do not expect it to have any influence on the pharmacokinetics of Linezolid. (lines 370-371)

This study was carried out after single dose without being in the steady state. We can see a higher V in obese patients versus non-obese patients. This data could provide information related to the loading dose required in this group of patients but not so much with the maintenance dose. In this sense I have to highlight:
1. Although the V is higher in obese group, it is similar to published data in other groups of patients while the V in non-obese ones is lower that was has been published.

An interesting observation, though the difference in V between obese and non-obese can be estimated well even if the absolute value for V, e.g. the intercept in a statistical model, is not precise.

Note as well that other studies in non-obese, non-critically ill patients showed comparable estimates of V (26.7-31.9 L, McGee et al. Population pharmacokinetics of linezolid in adults with pulmonary tuberculosis. Antimicrob Agents Chemother 2009) (lines 307-308).

 Regarding volume of distribution, it is important to mention that linezolid as a small and moderately lipophilic drug shows high permeability, distributing into the total body water (TBW). TBW is given in one study as 31.4±2.8 L for non-obese patients and 44.3±7.0 L for obese patient (Waki M et al. Relative expansion of extracellular fluid in obese vs. nonobese women. Am J Physiol Metab 1991), which agrees well with our values (30.8 L, 95% CI 27.9 L, 35.5 L for non-obese patients and 41.0 L 95% CI 38.4 L, 57.9 L for obese patients).

It can be hypothesized that an extension of total body water, e.g. via oedema, frequently occurring in critically ill patients, might contribute to a larger volume of distribution (Swoboda S et al. Pharmacokinetics of linezolid in septic patients with and without extended dialysis. Eur J Clin Pharmacol 2010).

2. The maintenance dose is more related to the clearance of linezolid.

Your distinction between loading and maintenance dose is important and has been included in the discussion (lines 342-352). We agree that our data are not sufficient for concrete dosage recommendations and refrain from providing any. Instead we intend to highlight the PK differences in obese and non-obese patients which should be further evaluated in the relevant patient population of critically ill obese patient as outlined in the Conclusions section.

3. There is no information in the text about patients’ renal function. An adequate estimation of renal function by calculating creatinine clearance measured in urine is critical to interpreting linezolid clearance.

We have added creatinine to Table 1. Urine creatinine has not been evaluated in this study. Creatinine clearance calculation based on serum creatinine has not yet been validated and there is an ongoing debate on which body size descriptor (total body weight, lean body weight, etc) to include in the Cockcroft-Gault formula to obtain the least biased estimate of creatinine clearance (Winter MA et al. Impact of various body weights and serum creatinine concentrations on the bias and accuracy of the cockcroft-gault equation. Pharmacotherapy 2012; Demirovic JA et al. Estimation of creatinine clearance in morbidly obese patients. Am J Heal Pharm 2009).

4. In this study linezolid is used as a single dose in prophylaxis. I understand that's why the maintenance dose hasn't been studied. But the usual use of linezolid is not in this context. It would be interesting to analyze the factors that may condition the clearance of linezolid in order to estimate the necessary maintenance dose in this group. Why has a non-compartmental model been selected instead of a compartmental one? Justify.

NCA was deemed adequate given the following prerequisites as recommended by the EMA guideline on the investigation of bioequivalence:

  • Frequent sampling around (predicted) tmax, i.e. end of infusion, to cover Cmax
  • At least three to four samples (compare Fig. 3, main manuscript) in the terminal log-linear phase in order to reliably estimate the terminal rate constant (to reliably estimate AUC0-∞)

Moreover, our study data originated from a single centre and hence there was no need to account for inter-occasion variability.

Between-patient variability was approximated using a linear mixed-effects modelling approach, based on a random effect term on the slope of the terminal phase.

We regard nonlinear mixed-effects modelling as an appropriate tool to further investigate alternative dosing regimens (which we allude to in our Discussion).

  • Instead of estimating the AUC/MIC value for each patient, a better approach would be to calculate the probability of target  attainment  using Monte Carlo simulation. This issue should be considered.

To the extent in which two groups of 15 patients allow for such analyses, they have been performed in the second paper mentioned in the discussion, which includes probability of target  attainment  using Monte Carlo simulation. As mentioned in our response to Reviewer 1, the choice to present the current analysis separately from the nonlinear mixed effects model with compartmental methods and Monte-Carlo (MC) simulations was intended to distinguish clearly between the foreseen protocol analysis and a post-hoc exploratory one. Moreover, the detailed description of the latter would have exceeded the scope of this paper.

Round 2

Reviewer 1 Report

No further comments.